# Efficiently Factorizing Boolean Matrices using Proximal Gradient Descent

**Sebastian Dalleiger**
CISPA Helmholtz Center for Information Security
sebastian.dalleiger@cispa.de

**Jilles Vreeken**
CISPA Helmholtz Center for Information Security
jv@cispa.de

## Abstract

Addressing the interpretability problem of NMF on Boolean data, Boolean Matrix Factorization (BMF) uses Boolean algebra to decompose the input into low-rank Boolean factor matrices. These matrices are highly interpretable and very useful in practice, but they come at the high computational cost of solving an NP-hard combinatorial optimization problem. To reduce the computational burden, we propose to relax BMF continuously using a novel elastic-binary regularizer, from which we derive a proximal gradient algorithm. Through an extensive set of experiments, we demonstrate that our method works well in practice: On synthetic data, we show that it converges quickly, recovers the ground truth precisely, and estimates the simulated rank exactly. On real-world data, we improve upon the state of the art in recall, loss, and runtime, and a case study from the medical domain confirms that our results are easily interpretable and semantically meaningful.

## 1 Introduction

Discovering groups in data and expressing them in terms of common concepts is a central problem in many scientific domains and business applications, including cancer genomics [23], neuroscience [13], and recommender systems [18]. This problem is often addressed using variants of *matrix factorization*, a family of methods that decompose the target matrix into a set of typically low-rank factor matrices whose product approximates the input well. Prominent examples of matrix factorization are Singular Value Decomposition (SVD) [10], Principal Component Analysis (PCA) [10], and Nonnegative Matrix Factorization (NMF) [31, 21, 22]. These methods differ in how they constrain the matrices involved: SVD and PCA require orthogonal factors, while NMF constrains the target matrix and the factors to be nonnegative.

SVD, PCA, and NMF achieve interpretable results—unless the data is Boolean, which is ubiquitous in the real world. In this case, their results are hard to interpret directly because the input domain differs from the output domain, such that post-processing is required to extract useful information. *Boolean Matrix Factorization* (BMF) addresses this problem by seeking two low-rank *Boolean* factor matrices whose Boolean product is close to the Boolean target matrix [24]. The output matrices, now lying in the same domain as the input, are interpretable and useful, but they come at the computational cost of solving an NP-hard combinatorial optimization problem [30, 24, 25]. To make BMF applicable in practice, we need efficient approximation algorithms.

There are many ways to approximate BMF—for example, by exploiting its underlying combinatorial or spatial structure [24, 6, 5], using probabilistic inference [37, 35, 36], or solving the related Bi-Clustering problem [28, 29]. Although these approaches achieve impressive results, they fall short when the input data is large and noisy. Hence, we take a different approach to overcome BMF's computational barrier. Starting from an NMF-like optimization problem, we derive a continuous relaxation of the original BMF formulation that allows intermediate solutions to be real-valued. Inspired by the elastic-net regularizer [43], we introduce the novel *elastic binary (ELB) regularizer* to

36th Conference on Neural Information Processing Systems (NeurIPS 2022).

regularize towards Boolean factor matrices. We obtain an efficient-to-compute *proximal operator* from our ELB regularizer that projects relaxed real-valued factors towards being Boolean, which allows us to leverage fast gradient-based optimization procedures. In stark contrast to the state of the art [15, 16, 17], which requires heavy post-hoc post-processing to actually achieve Boolean factors, we ensure a Boolean outcome upon convergence by gradually increasing the projection strength using a *regularization rate*. We combine our relaxation, efficient proximal operator, and regularization rate into an *Elastic Boolean Matrix Factorization* algorithm (ELBMF) that scales to large data, results in accurate reconstructions, and does so without relying on heavy post-processing procedures. ELB and its rate are, however, not confined to BMF and can regularize, e.g., binary MF or bi-clustering [17].

In summary, our main contributions are as follows:

1. We introduce the ELB regularizer.
2. We overcome the computational hardness of BMF leveraging a novel relaxed BMF problem.
3. We efficiently solve the relaxed BMF problem using an optimization algorithm based on proximal gradient descent.

The remainder of the paper proceeds as follows. In Sec. 2, we formally introduce the BMF problem and its relaxation, define our ELB regularizer and its proximal point operator, and show how to ensure a Boolean outcome upon convergence. We discuss related work in Sec. 3, validate our method through an extensive set of experiments in Sec. 4, and conclude with a discussion in Sec. 5.

## 2    Theory

Our goal is to factorize a given Boolean target matrix into at least two smaller, low-rank Boolean factor matrices, whose product comes close to the target matrix. Since the factor matrices are Boolean, this product follows the algebra of a Boolean semi-ring, i.e., it is identical to the standard outer product on a field where addition obeys $1 + 1 = 1$. We define the product between two Boolean matrices $U \in \{0, 1\}^{n \times k}$ and $V \in \{0, 1\}^{k \times m}$ on a Boolean semi-ring $(\{0, 1\}, \vee, \wedge)$ as

$$[U \circ V]_{ij} = \bigvee_{l \in [k]} U_{il} V_{lk} , \tag{1}$$

where $U \in \{0, 1\}^{n \times k}$, $V \in \{0, 1\}^{k \times m}$, and $U \circ V \in \{0, 1\}^{n \times m}$. This gives rise to the BMF problem.

**Problem 1 (Boolean Matrix Factorization)** *For a given target matrix* $A \in \{0, 1\}^{n \times m}$*, a given matrix rank* $\mathbb{N} \ni k \leq \min\{n, m\}$*, and* $A \oplus B$ *denoting logical exclusive or, discover the factor matrices* $U \in \{0, 1\}^{n \times k}$ *and* $V \in \{0, 1\}^{k \times m}$ *that minimize*

$$\|A - U \circ V\|_F^2 = \sum_{ij} A_{ij} \oplus [U \circ V]_{ij} . \tag{2}$$

While beautiful in theory, this problem is NP-complete [25]. Thus, we cannot solve this problem exactly for all but the smallest matrices. In practice, we hence have to rely on approximations. Here, we relax the Boolean constraints of Eq. (2) to allow non-negative, *non-Boolean* 'intermediate' factor matrices during the optimization, allowing us to use linear algebra rather than Boolean algebra. In other words, we solve the non-negative matrix factorization (NMF) problem [31]

$$\|A - UV\|_F^2 , \tag{3}$$

subject to $U \in \mathbb{R}_+^{n \times k}$ and $V \in \mathbb{R}_+^{k \times m}$. In contrast to the original BMF formulation, we can solve this problem efficiently, e.g., via a Gauss-Seidel scheme. Although efficient, using plain NMF, however, disregards the Boolean structure of our matrices and produces factor matrices from a different domain, which are consequently hard to interpret and potentially very dense. To benefit from efficient optimization and still arrive at Boolean outputs, we allow real-valued intermediate solutions and regularize them towards becoming Boolean.

To steer our optimization towards Boolean solutions, we penalize non-Boolean solutions using a regularizer. This idea has been explored in prior work. There exists the $l_1$-inspired PRIMP

regularizer [15], which is $-\kappa[-|1 - 2x| + 1]$ for values inside $[0, 1]$ and $\infty$ otherwise, and the $l_2$-inspired bowl-shaped regularizer [42], which is $\lambda(x^2 - x)^2/2$ everywhere on the real line. Although both have been successfully applied to BMF, both also have undesirable properties: The PRIMP regularizer penalizes well *inside* the interval $[0, 1]$ but is non-differentiable on the outside, while the bowl-shaped regularizer is differentiable and penalizes well *outside* the interval $[0, 1]$ but is almost flat on the inside. Hence, both regularizers are problematic if used individually. Combining them, however, yields a regularizer that penalizes non-Boolean values well across the full real line. To combine $l_1$- and $l_2$-regularization, we use the *elastic-net regularizer*,

$$r(x) = \kappa\|x\|_1 + \lambda\|x\|_2^2 ,$$

which, however, only penalizes *non-zero* solutions [43]. To penalize *non-Boolean* solutions, we combine two elastic-net regularizers into our (almost W-shaped) ELB *regularizer*,

$$R(X) = \sum_{x \in X} \min\{r(x), r(x - 1)\} , \tag{4}$$

where $X \in \{\mathrm{U}, \mathrm{V}\}$. In Fig. 1, we show all three regularizers in the range of $[-1, 2]$, for $\lambda = \kappa = 0.5$. We see that only the ELB regularizer penalizes non-Boolean solutions across the full spectrum, and summarize our regularized relaxed BMF as follows.

**Problem 2 (Elastic Boolean Matrix Factorization)** *For a given target matrix* $\mathrm{A} \in \{0, 1\}^{n \times m}$ *and a given matrix rank* $\mathbb{N} \ni k \leq \min\{n, m\}$, *discover the factor matrices* $\mathrm{U} \in \mathbb{R}_+^{n \times k}$ *and* $\mathrm{V} \in \mathbb{R}_+^{k \times m}$ *that minimize*

$$\|\mathrm{A} - \mathrm{UV}\|_F^2 + R(\mathrm{U}) + R(\mathrm{V}) . \tag{5}$$

Although this is a relaxed problem, it is still non-convex, and therefore, we cannot solve it straightforwardly. The problem, however, is suitable for the Gauss-Seidel optimization scheme. That is, we alternatingly fix one factor matrix to optimize the other. By doing so, we generate a sequence

$$\mathrm{U}_{t+1} \leftarrow \arg\min_{\mathrm{U}} \|\mathrm{A} - \mathrm{UV}_t\|_F^2 + R(\mathrm{U}) , \tag{6}$$

$$\mathrm{V}_{t+1} \leftarrow \arg\min_{\mathrm{V}} \|\mathrm{A} - \mathrm{U}_{t+1}\mathrm{V}\|_F^2 + R(\mathrm{V}) , \tag{7}$$

of simpler-to-solve sub-problems, until convergence. Now, each sub-problem is again a sum of two $f(X) + R(X)$ functions, where $f$ is the loss $\|\cdot\|_F^2$, and $R(X)$ is the regularizer. This allows us to follow a proximal gradient approach, i.e., we use *Proximal Alternating Linear Minimization* (PALM) [7, 33]. In a nutshell, we minimize a sub-problem by following the gradient $\nabla f$ of $f$, to then use the proximal operator for $R$ to nudge its outcome towards a Boolean solution. That is, for the gradients $\nabla_{\mathrm{U}} f = \mathrm{UVV}^\top - \mathrm{AV}^\top$ and $\nabla_{\mathrm{V}} f = \mathrm{U}^\top \mathrm{UV} - \mathrm{U}^\top \mathrm{A}$, we compute the step

$$\mathrm{prox}_R(X - \eta\nabla f) , \tag{8}$$

where $\eta$ represents the step size, which we compute in terms of Lipschitz constant, rather than relying on a costly line-search [7]. To further improve the convergence properties, we make use of an inertial term that linearly combines $X_t$ with $X_{t-1}$ before applying Eq. (8) (see [33] for a detailed description). We now derive the proximal operator for the ELB regularizer, before discussing how we ensure that the factor matrices are Boolean, and summarizing our approach as an algorithm.

## 2.1 Proximal Mapping

To solve the sub-problem

$$\arg\min_X \ell_{\kappa\lambda}(X) \text{ for } \ell_{\kappa\lambda}(X) = f(X) + R(X) \tag{9}$$

from Eq. (7) for $X \in \{\mathrm{U}, \mathrm{V}\}$, we need a proximal operator [32] that projects values towards a regularized point. Formalized in Appendix A, in a nutshell, this operator is the solution to

$$\mathrm{prox}_R(X) = \arg\min_Y \frac{1}{2}\|X - Y\|_2^2 + R(X) . \tag{10}$$

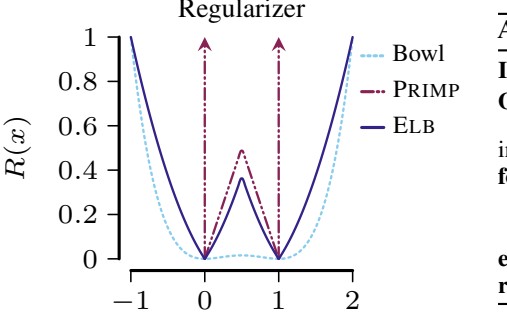

| Algorithm 1: ELBMF |
| --- |
| **Input:** Matrix $A \in \{0,1\}^{n \times m}$, rank $k \in \mathbb{N}$ |
| **Output:** Factors $U \in \{0,1\}^{n \times k}$, $V \in \{0,1\}^{k \times m}$ |
| initialize U and V uniformly at random |
| **for** $t = 1, 2, \ldots$ **until convergence do** |
| $\quad U \leftarrow \arg \text{reduce}_U \ell_{\kappa \lambda_t}(A, U, V)$ |
| $\quad V \leftarrow \arg \text{reduce}_V \ell_{\kappa \lambda_t}(A, U, V)$ |
| **end** |
| **return** $U, V$ |

Figure 1: On the left, we show the three regularizers: Bowl, PRIMP, and ELB, for $\lambda = \kappa = 0.5$, and see that only our ELB regularizer penalizes non-Boolean values well. On the right, we show our method ELBMF as pseudocode.

As Eq. (10) is coordinate-wise solvable, we can reduce this equation to a scalar proximal operator

$$\text{prox}_{\kappa\lambda}(x) = \underset{y \in \mathbb{R}}{\arg\min} \frac{1}{2}(x-y)^2 + R(y) . \tag{11}$$

Although this scalar function may seem simple, due to $R$, it is non-convex, which—in general—rules out the existence of unique minima. We can, however, exploit the W-shape of $R$. That is, from the perspective of a fixed $X_{ij}$, the regularizer is locally convex after a case distinction. Starting with the case of $X_{ij} = x$, $x \leq \frac{1}{2}$, we can simplify Eq. (11) to $\frac{1}{2}(x-y)^2 + \lambda'/2y^2 + \kappa|y|$, letting $\lambda' = 2\lambda$. Setting its derivative to 0, we get to $y = x - \kappa \text{sign}(y)$. Asserting that a least-squares solution will always be of the same sign, we substitute the sign of $x$ with $\text{sign}(y)$. Repeating these steps analogously for $x > 1/2$, we obtain our proximal operator:

**Definition 1 (Proximal Operator)** *Given regularization coefficients $\kappa$ and $\lambda$, our proximal operator for matrix $X$ is $\text{prox}_R(X) = [\text{prox}_{\kappa\lambda}(X_{ij})]_{ij}$, where $\text{prox}_{\kappa\lambda}$ is the scalar proximal operator*

$$\text{prox}_{\kappa\lambda}(x) \equiv (1+\lambda)^{-1} \begin{cases} x - \kappa \, \text{sign}(x) & \text{if } x \leq \frac{1}{2} \\ x - \kappa \, \text{sign}(x-1) + \lambda & \text{otherwise .} \end{cases} \tag{12}$$

Although not strictly necessary, to improve the empirical convergence rate, we would like to constrain our factor matrices to be non-negative. As our proximal operator does not account for this, we impose non-negativity by using the *alternating projection* procedure to combine the non-negativity proximal operator [32] with Eq. (12) into $\max\{0, \text{prox}_{\kappa\lambda}(x)\}$.

## 2.2 Ensuring Boolean Factors

Our proximal operator only nudges the factor matrices towards *becoming* Boolean. We, however, want to ensure that our results *are* Boolean. To this end, the state-of-the-art method PRIMP relies heavily on post-processing, performing a very expensive joint two-dimensional grid search to guess the 'best' pair of rounding thresholds, which are then used to produce Boolean matrices. Although this tends to work in practice, it is an inefficient post-hoc procedure—and thus, it would be highly desirable to have Boolean factors already upon convergence. To achieve this without rounding or clamping, we revisit our regularizer, which binarizes more strongly if we regularize more aggressively. Consequently, if we regularize too aggressively, we converge to a suboptimal solution, and if we regularize too mildly, we do not binarize our solutions. To prevent subpar solutions and still binarize our output, we start with a weak regularization and gradually increase its strength.

Considering Eq. (12), we see that a higher regularization strength increases the distance over which our proximal operator projects. Thus, if we set the $l_1$-distance controlling $\kappa$ too high, we will immediately leap to a Boolean factor matrix, which will terminate the algorithm and yield a suboptimal solution.

Regulating the $l_2$-distance controlling $\lambda$ is a less delicate matter. Hence, we gradually increase $\lambda$ to prevent a subpar solution and achieve a Boolean outcome, using a *regularization rate*

$$\lambda_t = \lambda \cdot \nu_t \quad \text{for} \quad \nu_t \geq 0 \quad \forall t \geq 0 \tag{13}$$

that gradually increases the proximal distance at a user-defined rate. In case ELBMF has stopped without convergence, we bridge the remaining integrality gap by projecting the outcome onto its closed Boolean counterpart, using our proximal operator (see Fig. 2).

We summarize the considerations laid out above as ELBMF in Alg. 1. The computational complexity of ELBMF is bounded by the complexity of computing the gradient, which is identical to the complexity of matrix multiplication. Therefore, for all practical purposes, ELBMF is sub-cubic $\mathcal{O}\left(n^{2.807}\right)$ using Strassen's algorithm.

## 3   Related Work

*Matrix factorization* is a well-established family of methods, whose members, such as SVD, PCA, or NMF, are used everywhere in machine learning. Almost all matrix factorization methods operate on real-valued matrices, however, while BMF operates under Boolean algebra. *Boolean Matrix Factorization* originated in the combinatorics community [27] and was later introduced to the data mining community [24], where many cover-based BMF algorithms were developed [6, 5, 24, 26]. In recent years, BMF has gained traction in the machine learning community, which tends to tackle the problem differently. Here, *relaxation-based approaches* that optimize for a relaxed but regularized BMF [15, 14, 42] are related to our method, but they differ especially in their regularization. Hess et al. [15] introduce a regularizer that is only partially differentiable, and they rely heavily on post-processing to force a Boolean solution, and Zhang et al. [42] regularize only weakly between 0 and 1. In contrast, our regularizer penalizes well across the full spectrum and yields a Boolean outcome upon convergence. Building on a thresholding-based BMF formulation, Araujo et al. [3] also consider relaxations to benefit from gradient-based optimization. Other recent approaches build on *probabilistic inference*. Rukat et al. [37, 35, 36], for example, combine Bayesian Modeling and sampling into their logical factor machine. A similar direction is taken by Ravanbakhsh et al. [34], who use graphical models and message passing, and Liang et al. [23], who combine MAP-inference and sampling. A different, *geometry-based approach* lies in locating dense submatrices by ordering the data to exploit the consecutive-ones property [40, 38]. Since BMF is essentially solving a bipartite graph partitioning problem, it is also closely related to Bi-Clustering and Co-Clustering [28, 17]. Neumann and Miettinen [29] use this relationship to efficiently solve BMF by means of a streaming algorithm. Although there are many different approaches to BMF, its biggest challenge to date remains scalability [25].

## 4   Experiments

We implement ELBMF in the Julia language and run experiments on 16 Cores of an AMD EPYC 7702 and a single NVIDIA A100 GPU, reporting wall-clock time. We provide the source code, datasets, synthetic dataset generator, and other information needed for reproducibility.[1] We compare ELBMF against six methods: four dedicated BMF methods (ASSO [24], GRECOND [6], ORM [35], and PRIMP [15]), one streaming Bi-Clustering algorithm SOFA [29], one elastic-net-regularized NMF method leveraging proximal gradient descent (NMF [31, 21, 22]), and one interpretable Boolean autoencoder (BINAPS [9]). Since NMF outputs non-negative factor matrices, rather Boolean matrices, we cannot compare against NMF directly, so we clamp and round its solutions to the nearest Boolean outcome. To fairly compare against BINAPS, we task it with autoencoding the target matrix as a reconstruction, given the matrix ranks from our experiments as the number of latent dimensions. We perform three sets of experiments. First, we ascertain that ELBMF works reliably on synthetic data. Second, we verify that it generally performs well on real-world data. And third, we illustrate that its outputs are semantically meaningful through an exploratory analysis of a biomedical dataset.

### 4.1   Performance of ELBMF on Synthetic Data

In the following experiments, we ask four questions: (**1**) How does ELBMF converge?; (**2**) How well does ELBMF recover the information in the target matrix?; (**3**) How consistently does ELBMF

---

[1]  Appendix C; DOI: 10.5281/zenodo.7187021

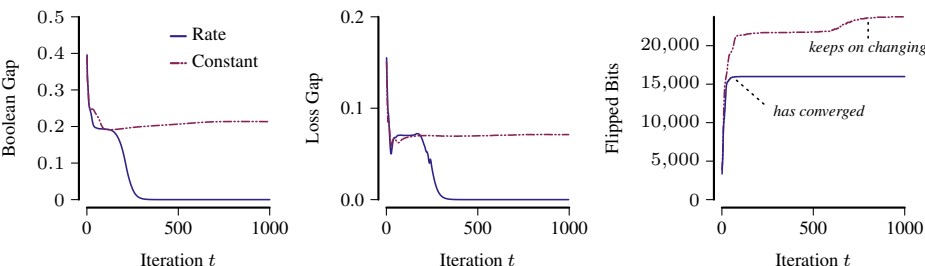

Figure 2: Our method converges quickly under a regularization rate. We report the progression over time of the *Boolean gap*, *loss gap*, and the *Hamming process*, for $1\,000$ iterations of ELBMF on synthetic $400 \times 300$ matrices with $10\%$ noise and 5 random tiles covering between 50 and 100 rows or columns, under a constant regularization of $\lambda_t = 1$ or a regularization rate of $\lambda_t = 1.05^t$.

reconstruct low- or high-density target matrices?; and (**4**) Does ELBMF estimate the underlying Boolean matrix rank correctly? To answer these questions, we generate synthetic data with known ground truth as follows. Starting with an all-zeros matrix, we randomly create rectangular, non-overlapping, consecutive areas of ones called *tiles*, each spanning a randomly chosen number of consecutive rows and columns, thus inducing matrices with varying densities. We then add noise by setting each cell to 1, uniformly at random, with varying noise probabilities.

**How does ELBMF converge?**    To study how our method converges to a Boolean solution, we quantify relevant properties of the sequence of intermediate solutions (cf. Eq. (7)). First, to understand how quickly and stably ELBMF converges to a Boolean solution, we quantify the *Boolean gap*,

$$\sum_{X \in \{U_t, V_t\}} |X|^{-1} \sum_{x \in X} \min\{|x|, |x - 1|\} \ .$$

Second, to understand when we can safely round intermediate almost-Boolean solutions without losing information, we calculate, for the reconstruction B from *rounded* intermediate factors, the cumulative *Hamming process* as the sum of fraction of bits that flip from iteration $t$ to iteration $t + 1$,

$$|A|^{-1}\|\mathrm{B}_t - \mathrm{B}_{t+1}\|_1 \ ,$$

and the *loss gap* as the difference between the relaxed loss and the loss from the *rounded* B.

As shown in Fig. 2, we achieve an almost-Boolean solution *without any rounding* after around 250 epochs, continuing until we reach a Boolean outcome. This is also the point at which the rounded intermediate solution and its relaxation are almost identical, as illustrated by the loss gap. Considering the Hamming process on the right, we observe that ELBMF goes through an erratic bit-flipping-phase in the beginning, followed by only minor changes in each iteration until iteration $t = 100$. Afterwards, ELBMF has settled on a solution—under our regularization rate. When using constant regularization instead, we continue to observe bit flips until the end of the experiment. Under constant regularization, the Boolean gap hardly decreases over time. Far from Boolean, the constant regularization thus also never closes the loss gap, which is unsurprising, given that its factors are less regularized. In other words, our regularization works well, and it allows us to safely binarize almost-converged factors that are $\epsilon$-far from being Boolean by means of, e.g., our proximal operator.

**How well does ELBMF recover the information in the target matrix?**    Having ensured that our method converges stably and quickly, we would like to assess whether it also converges to a high-quality factorization. To this end, we generate synthetic $40 \times 30$ matrices containing 5 random tiles each spanning 5 to 10 rows and columns, under additive noise levels between $0\%$ (no noise) and $50\%$. We then compute the fraction of *ones* in target A that is covered by the reconstruction $B = U \circ V$, i.e., the *recall* (higher is better)

$$\|\mathrm{A}\|_1^{-1}\|\mathrm{A} \odot \mathrm{B}\|_1 \ . \tag{14}$$

To ensure that we fit the *signal* in the data, we additionally report the recall regarding the generating, noise-free ground-truth tiles $A^*$, denoted as recall$^*$. Finally, to rate the overall reconstruction quality

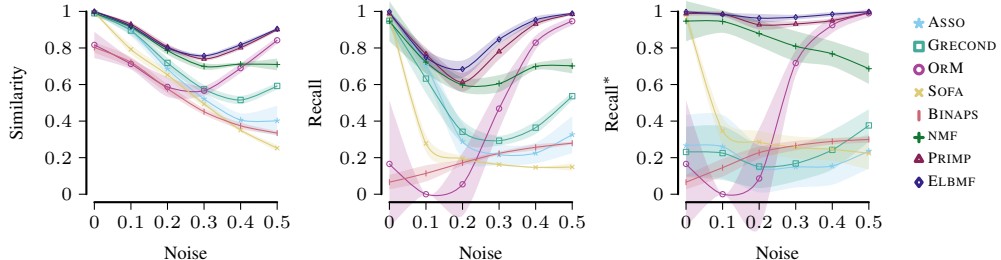

Figure 3: Overall, ELBMF reconstructs the noisy synthetic data well and recovers the ground-truth tiles. On synthetic data for additive noise levels increasing from $0\%$ to $50\%$, we show mean as line and standard deviation as shade of *similarity*, *recall* w.r.t. the target matrix, and *recall** w.r.t. the noise-free ground-truth tiles, for ASSO, GRECOND, ORM, SOFA, BINAPS, NMF, PRIMP, and ELBMF.

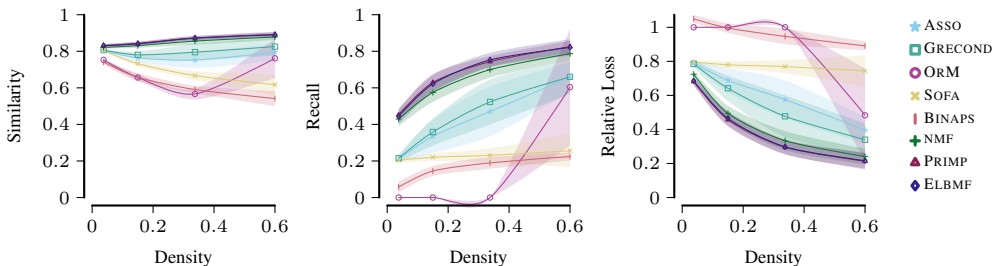

Figure 4: ELBMF reconstructs noisy synthetic high-density and low-density matrices consistently well. On synthetic data with fixed additive noise, and increasing density, we show mean as line and standard deviation as shade of *similarity*, *recall*, and *relative loss*, for BINAPS, ASSO, GRECOND, ORM, SOFA, BINAPS, NMF, PRIMP, and ELBMF.

including *zeros*, we compute the *Hamming similarity* (higher is better) between the target matrix and its reconstruction

$$|A|^{-1}\|A - B\|_1 \;.$$

We run each method on our synthetic datasets, targeting a matrix rank of 5. To account for random fluctuations, we average over 10 randomly drawn sets of 5 ground-truth tiles per $10\%$ increment in noise probability. In Fig. 3, we show similarity, recall, and recall*. We observe that in the noiseless case ($0\%$), all methods except BINAPS recover the 5 ground-truth tiles with high accuracy, but only ASSO and ELBMF do so with perfect recalls. Starting with as little as $10\%$ noise, both recalls of ASSO, GRECOND, SOFA, NMF, and BINAPS deteriorate quickly, while the similarity and both recalls of PRIMP and ELBMF remain high. In fact, ELBMF and PRIMP perform similarly across the board—which is highly encouraging, as unlike PRIMP, ELBMF does not require post-processing. For ASSO and GRECOND, recall and similarity drop considerably, but they exhibit a slightly higher recall*. This means that these methods are robust against noise, but they fail to recover the remaining information. Starting low, ORM's recalls increase jointly with the noise level, suggesting that clean data is problematic for ORM. Reporting the standard deviations as the shaded region, we see little variance across all similarities—except for ASSO and NMF in the highest noise regime. The deviation of both recalls is, however, inconsistent for most methods, except for BINAPS, PRIMP, and ELBMF. Overall, the performance characteristics of ELBMF are among the most reliable.

**How robust is ELBMF regarding varying matrix densities?** To understand whether ELBMF performs consistently well on low- or high-density matrices, we generate synthetic matrices as before, this time using fixed noise of $0.2$, and varying the width and height of the ground-truth squared tiles from $3^2$ to $12^2$, resulting in densities between $0.0375$ to $0.6$, before noise.

In Fig. 4, we show the similarity, recall, and loss of BINAPS, ASSO, GRECOND, ORM, SOFA, NMF, PRIMP, and ELBMF. We can see that the increasing density affects the performance of all

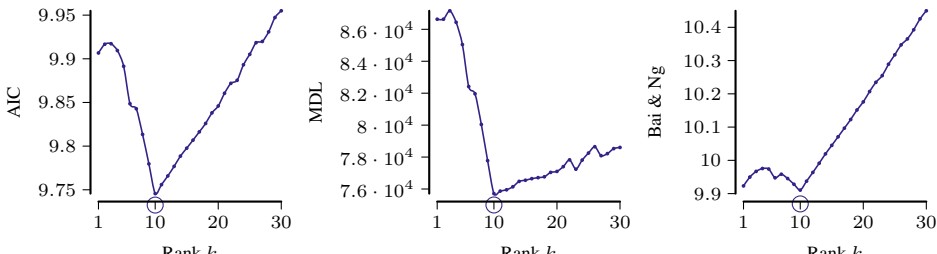

Figure 5: Using AIC, MDL, or Bai & Ng's first information criteria, ELBMF correctly detects the simulated $400 \times 300$ matrix of rank $10$ to which we applied $10\%$ additive noise.

methods, however, it does not affect the performance of all methods *equally*. All methods—except ORM—improve in similarity, recall, and loss. With increasing density, ORM gets worse at first, before its loss shrinks significantly, such that it finishes outperforming SOFA and BINAPS. From low to high density, ELBMF is the best-performing method across the board in similarity, recall, and loss.

**Does ELBMF estimate the underlying Boolean matrix rank correctly?**    When a target rank is known, we can immediately apply ELBMF to factorize the data. In the real world, however, the target rank might be unknown. In this case, we need to *estimate* an appropriate choice from the data, and we use synthetic data to ensure that ELBMF does so correctly.

Since a higher matrix rank usually also means a better fit, selecting the best rank according to recall, loss, or similarity leads to overfitting—unless we properly account for the growth in model complexity. There are many *model selection criteria* that penalize complex models, such as AIC [1], Bai & Ng's criteria [4, 2], Nuclear-norm regularizing [20, 12], the information-theoretic Minimal Description Length principle (MDL) [11], or (Decomposed) Normalized Maximum Likelihood [19, 41]. Following common practice, and motivated by its practical performance in preliminary experiments, we choose MDL. That is, we select the *minimizer* of the sum of the log binomial $l(X) = \log \binom{|X|}{\|X\|_1}$ of the error matrix and the rows and columns of our factorization (assumed to be i.i.d.) [26]

$$l(A \oplus [U \circ V]) + \sum_{i \in [k]} l(U_i^\top) + l(V_i) + k \log(n \cdot m) \ .$$

To validate whether ELBMF recovers the correct rank, we synthetically generate a $400 \times 300$ matrix of ground-truth rank 10 with $10\%$ additive noise. In Fig. 5, we show AIC, MDL, and Bai & Ng's first criterion for each rank up to 30, finding that our method precisely discovers the right rank.

## 4.2 Performance of ELBMF on Real-World Data

Having ascertained that ELBMF works well on synthetic data, we turn to its performance in the real world. Here, we use nine publicly available datasets[2] from different domains. To cover the *biomedical domain*, we extract the network containing empirical evidence of protein-protein interactions in *Homo sapiens* from the STRING database. From the GRAND repository, we take the gene regulatory networks sampled from *Glioblastoma (GBM)* and *Lower Grade Glioma (LGG)* brain cancer tissues, as well as from non-cancerous *Cerebellum* tissue. The *TCGA* dataset contains binarized gene expressions from cancer patients, and we further obtain the single nucleotide polymorphism (SNP) mutation data from the 1*k Genomes* project, following processing steps from the authors of BINAPS [9]. In the *entertainment domain*, we use the user-movie datasets *Movielens* and *Netflix*, binarizing the original 5-star-scale ratings by setting only reviews with more than $3.5$ stars to 1. Finally, as data from the *innovation domain*, we derive a directed citation network between patent groups from patent citation and classification data provided by *PatentsView*. For each dataset with a given number of groups, such as cancer types or movie genres, we set the matrix rank $k$ to 33 (TCGA), 28 (Genomes), 136 (Patents), 20 (Movielens), and 20 (Netflix). When the number of subgroups is unknown, we estimate the rank that minimizes MDL using ELBMF, resulting in 100 (GBM), 32 (LGG), 100 (String), and 450 (Cerebellum).

---

[2]  GRAND.NETWORKMEDICINE.ORG STRING-DB.ORG CANCER.GOV/TCGA INTERNATIONALGENOME.ORG PATENTSVIEW.ORG GROUPLENS.ORG/DATASETS/MOVIELENS  KAGGLE.COM/DATASETS/NETFLIX-INC/NETFLIX-PRIZE-DATA

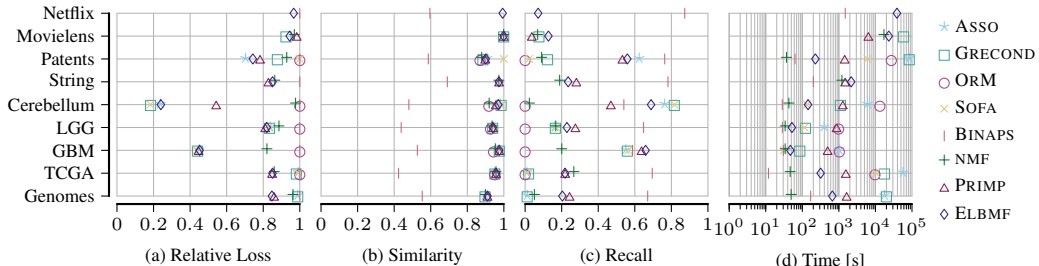

Figure 6: Our method factorizes real-world data with high similarity and recall, as well as low relative loss and runtime. We report *relative loss*, *similarity*, *recall*, and *runtime* of 9 real-world matrices and their reconstructions from ASSO, GRECOND, ORM, SOFA, BINAPS, NMF, PRIMP, and ELBMF.

As we can achieve a high similarity with an all-zeros reconstruction (in the case of sparse data), or a perfect recall with an all-ones reconstruction, we also report the *relative loss* (lower is better),

$$\|A\|_1 \|A - U \circ V\|_1 \;,$$

between the target matrices and their reconstructions.

We show relative loss, similarity, recall, and runtime of ASSO, GRECOND, SOFA, ORM, BINAPS, NMF, PRIMP, and ELBMF, applied to all real-world datasets, targeting a given matrix rank, in Fig. 6. The cover-based GRECOND and ASSO show comparable loss, similarity, and recall. Both perform better on smaller matrices (*LGG*, *GBM*, or *Cerebellum*) and struggle with complex matrices (e.g., *TCGA*, *Genomes*). On the complex matrices (e.g., on *TCGA*, *String*, or *Movielens*), although always outperformed by ELBMF, we see that the *rounded* NMF reconstructions are surprisingly good, occasionally surpassing dedicated BMF methods, such as ASSO, GRECOND, ORM, and SOFA. Across the board, GRECOND, ASSO, ORM, SOFA, BINAPS, and NMF almost always result in considerably higher loss than ELBMF. Compared to the close competitor PRIMP, our method ELBMF always results in lower reconstruction loss. We observe the largest gap between the two on the Cerebellum dataset, where PRIMP's grid search procedure fails to find suitable thresholds. This is an impressive result because unlike PRIMP, ELBMF does not necessitate heavy post-processing.

In Fig. 6b, we see that all methods except BINAPS result in a high similarity, which implies they are sparsity-inducing. As BINAPS overfits and densely reconstructs sparse inputs, it surpasses sparsity-inducing methods in recall. Considering non-overfitting methods, however, ELBMF is among the best performing in terms or recall, often outperforming PRIMP, while under significantly stronger regularization. When PRIMP has a higher recall (e.g., *Genomes*), this often comes with a higher loss than ELBMF. We see in Fig. 6d (log scale) that—except for few cases—ASSO, GRECOND, ORM, SOFA, and PRIMP are slower than ELBMF. Although NMF is less constrained than ELBMF, both are almost on par when it comes to runtime. Degraded by post-processing, our closed competitor PRIMP is almost always much slower than ELBMF and struggles with *Netflix*. Only BINAPS and ELBMF finished *Netflix*, however, only ELBMF did so at a reasonable loss, considering the given target rank.

## 4.3 Exploratory Analysis of Gene Expression Data with ELBMF

Knowing that ELBMF performs well quantitatively, we ask whether its outputs are also interpretable. To this end, we take a closer look at the *TCGA* data, which contains the expression levels of 20 530 genes from 10 459 patients, who are labeled with 33 cancer types. Since we are interested in retaining high gene expression levels only, we set expression levels to one if their $z$-scores fall into the top 5% quantile, and to zero otherwise [23]. We run ELBMF on this dataset, targeting a rank of 33.

To learn whether our method groups patients meaningfully, we visualize the target matrix and its reconstruction. As the target matrix is high-dimensional, we embed both the target matrix and the reconstruction into a two-dimensional space using t-SNE [39] as illustrated in Fig. 7, where each color corresponds to one cancer type. In Fig. 7a, we show that when embedding the target matrix directly, the cancer types are highly overlapping and hard to distinguish without the color coding. In contrast, when embedding our reconstruction, depicted in Fig. 7b, we see a clean segmentation into clusters that predominantly contain a single cancer type.

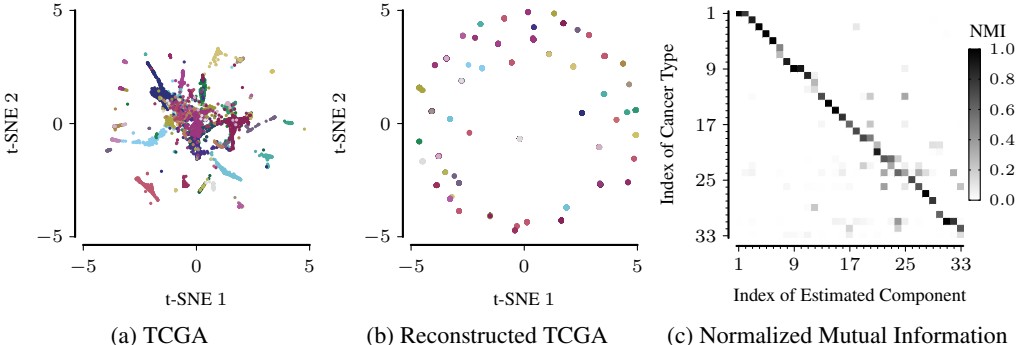

|  | (a) TCGA | (b) Reconstructed TCGA | (c) Normalized Mutual Information |

Figure 7: ELBMF discovers the hidden structure in gene expression data. We show the two-dimensional t-SNE embedding of the TCGA dataset (a) and the embedding of its reconstruction from ELBMF (b), where each point corresponds to one of the 10 459 patients, colored by cancer type. While the cancer types are hard to differentiate in the embedding of the original dataset (a), they are separated into easily distinguishable clusters in the embedding of our reconstruction (b). In (c), we show the normalized mutual information between estimated groups and cancer types, and observe that there is an almost 1-to-1 correspondence between our estimated groups and the cancer types.

To better understand these results, we quantify the association between our 33 estimated components and the ground-truth cancer types by computing the *normalized mutual information* matrix. This matrix is noticeably sparse (Fig. 7c), leading to a clean segmentation. Upon closer inspection with ENRICHR [8], the associations we discover turn out to be biologically meaningful. For example, we find that ELBMF associates a set of 356 genes to patients with *thyroid carcinoma*. This component is associated with *thyroid hormone generation* and *thyroid gland development*, and *statistically significantly* so—even under a strict False Discovery Control, with $p$-values as low as $2.574 \times 10^{-8}$ and $6.530 \times 10^{-6}$.

## 5   Conclusion

We introduced ELBMF to efficiently factorize Boolean matrices using an elegant and simple algorithm that, unlike its closest competitors, does not rely on heavy post-processing. ELBMF considers a novel relaxed BMF problem, which allows intermediate solutions to be real-valued. It solves this problem by leveraging an efficiently computable proximal operator, derived from the innovative ELB regularizer, and using a regularization rate to obtain Boolean factors upon convergence. Experimentally, we have shown that ELBMF works well in practice. It operates reliably on synthetic data, outperforms the discrete state of the art, is at least as good as the best relaxations on real data, and yields interpretable results even in difficult domains—without relying on post-processing.

**Limitations**   Although ELBMF works overall, it has two bottlenecks. First, by randomly initializing factors, we start with highly dense matrices, thus prohibiting efficient sparse matrix operations. This is not ideal for sparse datasets that are too large to fit into memory, and future research on sparse initialization of *Boolean* factors will benefit not only ELBMF but also many other methods. Second, the larger the datasets, the higher the cost of computing gradients, and future work might adapt stochastic gradient methods for ELBMF to mitigate this problem.

**Broader Impact**   ELBMF is a method for factorizing Boolean matrices, which permits interpretable, rather than black-box data analysis. As such, it can help make explicit any biases that may be present in the data. Just like any other method for interpretable data analysis, the insights ELBMF yields can be used for good purposes (e.g., insight into cancer-causing mutations) or for bad purposes, and the ultimate quality of the outcome also depends on the behavior of the user. In our experiments, we focus on beneficial applications, and only consider anonymized open data.

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
