## Appendix

We include the following supplementary materials in this Appendix.

In Appendix A, we formally derive our proximal operator from our ELB regularizer, resulting in the proximal operator Eq. (12) as well as an alternative proximal operator Eq. (17), not used in our experiments.

In Appendix B, we provide a more detailed version of Algorithm 1.

In Appendix C, we supply details regarding reproducibility, such as hyper-parameters, data-preprocessing, implementation details, and datasets.

In Appendix D, we share additional experiments on synthetic data under varying levels of additive noise, this time allowing for overlapping tiles.

In Appendix E, we provide additional experiments on synthetic data under a low additive noise and varying densities.

## A  Derivation of the Proximal Operator

In this section, we derive our proximal operator (given in Eq. (12), repeated below)

$$\text{prox}_{\kappa\lambda}(x) \equiv (1+\lambda)^{-1} \begin{cases} x - \kappa\,\text{sign}(x) & \text{if } x \leq \frac{1}{2} \\ x - \kappa\,\text{sign}(x-1) + \lambda & \text{otherwise}, \end{cases} \tag{15}$$

from the ELB (Eq. (4), repeated below)

$$R(X) = \sum_{x \in X} \min\{r(x), r(x-1)\}\,.$$

Starting with the definition [32] of the general proximal operator

$$\arg\min_{Y} \frac{1}{2}\|X - Y\|_2^2 + R(X)\,,$$

we observe that this proximal operator is coordinate-wise solvable. This allows us to derive a *scalar* proximal operator, which we then apply to each value in the matrix independently. Substituting the regularizer $R$ with its scalar version, we obtain a scalar proximal operator (Eq. (11))

$$\text{prox}_{\kappa\lambda}(x) = \arg\min_{y \in \mathbb{R}} \frac{1}{2}(x-y)^2 + \min\{r(y), r(y-1)\}\,,$$

which is non-convex, has no unique minima, and, therefore, is not straightforwardly solvable. We can, however, separate this function into two *locally convex* V-shaped functions, which are straightforwardly solvable. By asserting that its least-squares solution is either less than $1/2$ (if $x \leq 1/2$), or greater than $1/2$ (if $x > 1/2$), we can address each case independently, and merge the outcome into a single piecewise proximal operator.

**Case I**  In the first case, we address the operator for $x \leq 1/2$. For this, we start by simplifying our scalar proximal operator Eq. (11), by substituting $r(y)$ with its definition, and get

$$\text{prox}_{\kappa\lambda}^{\leq 1/2}(x) = \frac{1}{2}(y-x)^2 + \frac{\lambda'}{2}y^2 + \kappa|y|\,,$$

for $\lambda'/2 = \lambda$. Then, we take its partial derivative for $y$

$$\frac{\partial}{\partial y}\text{prox}_{\kappa\lambda}^{\leq 1/2}(x) = (y-x) + \lambda'y + \kappa\,\text{sign}(y)\,,$$

which we set to zero, obtaining

$$0 = (y-x) + \lambda'y + \kappa\,\text{sign}(y) \Leftrightarrow 0 = y(1+\lambda') - x + \kappa\,\text{sign}(y)\,.$$

By asserting that we can obtain a better least-squares solution if $y$ has the same sign as $x$, we can substitute the sign of $x$ with $\text{sign}(y)$, and get

$$0 = y(1+\lambda') - x + \kappa\,\text{sign}(x) \Leftrightarrow y = (1+\lambda')^{-1}[x - \kappa\,\text{sign}(x)]\,,$$

which concludes the first case.

**Case II**  Analogously, we now repeat the steps from above for $x > \text{\small 1}/\text{\small 2}$. Again, we start by simplifying Eq. (11), substituting $r(y-1)$

$$\text{prox}_{\kappa\lambda}^{> 1/2}(x) = \frac{1}{2}(y-x)^2 + \frac{\lambda'}{2}(y-1)^2 + \kappa|y-1| \,,$$

for $\lambda'/2 = \lambda$. By taking its partial derivative for $y$

$$\frac{\partial}{\partial y}\text{prox}_{\kappa\lambda}^{> 1/2}(x) = (y-x) + \lambda'(y-1) + \kappa\,\text{sign}(y-1) \,,$$

and setting it to zero, we obtain

$$0 = (y-x) + \lambda'(y-1) + \kappa\,\text{sign}(y-1) \Leftrightarrow 0 = (1+\lambda')y - \lambda' - x + \kappa\,\text{sign}(y-1) \,.$$

Then, asserting that the least-squares solution does not get worse by using the same sign for $y-1$ and $x-1$, we can substitute the sign of $x-1$ with $\text{sign}(y-1)$, and get

$$0 = y(1+\lambda') - x - \lambda' + \kappa\,\text{sign}(x-1) \Leftrightarrow y = (1+\lambda')^{-1}[x - \kappa\,\text{sign}(x-1) + \lambda'] \,,$$

which concludes the $x > \text{\small 1}/\text{\small 2}$ case.

**Combining Case I & Case II**  Combining the cases above yields our piecewise proximal operator (see Eq. (12))

$$\text{prox}_{\kappa\lambda}(x) \equiv (1+\lambda)^{-1} \begin{cases} x - \kappa\,\text{sign}(x) & \text{if } x \leq \frac{1}{2} \\ x - \kappa\,\text{sign}(x-1) + \lambda & \text{otherwise} \,. \end{cases} \tag{16}$$

**Alternative Proximal Operator**  Considering Eq. (11), we notice that the term $y-x$ is squared, which means that there are multiple solutions to this equation. We derive the alternative operator analogously to the steps above, however, by switching the positions of $y$ and $x$ in $f$.

$$\text{prox}_{\kappa\lambda}^{\text{alt.}}(x) \equiv (\lambda-1)^{-1} \begin{cases} -x - \kappa\,\text{sign}(x) & \text{if } x \leq \frac{1}{2} \\ -x - \kappa\,\text{sign}(x-1) + \lambda & \text{otherwise} \,. \end{cases} \tag{17}$$

Since this operator is denominated by $\lambda' - 1$, we need to ensure that $\lambda' \neq 1$. Because our original proximal operator in Eq. (12) is denominated by $\lambda' + 1$, and since $\lambda'$ is usually positive, we are not required to take extra precautions. Since this is more convenient, we select Eq. (12) as our proximal operator, rather than taking extra precautions when using Eq. (17).

# B   Extended pseudocode for ELBMF

Extending Alg. 1, we provide the more detailed version of ELBMF as pseudocode in Alg. 2.

---

**Algorithm 2: Long version of ELBMF in terms of iPALM [33]**

---

|         |                              |                                       |
|---------|------------------------------|---------------------------------------|
|         | Target Matrix                | $A \in \{0,1\}^{n \times m}$          |
|         | Rank                         | $k \in \mathbb{N}$,                   |
|         | $l_1$ Regularizer Coefficients | $\kappa \in \mathbb{R}$,            |
| **Input:** | $l_2$ Regularizer Coefficients | $\lambda \in \mathbb{R}$,           |
|         | Regularization Rate          | $\nu_t \in \mathbb{N} \to \mathbb{R}$,|
|         | **optional** Inertial Parameter | $\beta \in \mathbb{R}_+$           |

**Output:** Factors $U \in \{0,1\}^{n \times k}$, $V \in \{0,1\}^{k \times m}$

$U_0 = U_1 \leftarrow \mathrm{rand}(n, k)$
$V_0 = V_1 \leftarrow \mathrm{rand}(k, m)$

**for** $t = 1, 2, \ldots$ **until convergence do**
$\quad \lambda_t \leftarrow \lambda \cdot \nu_t$

$\quad U \leftarrow U_{t-1} + \beta(U_{t-1} - U_{t-2})$
$\quad \nabla_U f = UVV^\top - AV^\top$
$\quad L \leftarrow \|VV^\top\|_2$
$\quad U \leftarrow \mathrm{prox}_{\kappa L^{-1}, \lambda_t L^{-1}} \left( U - L^{-1} \nabla_U f \right)$
$\quad U_t \leftarrow U$

$\quad V \leftarrow V_{t-1} + \beta(V_{t-1} - V_{t-2})$
$\quad \nabla_V f = U^\top UV - U^\top A$
$\quad L \leftarrow \|U^\top U\|_2$
$\quad V \leftarrow \mathrm{prox}_{\kappa L^{-1}, \lambda_t L^{-1}} \left( V - L^{-1} \nabla_V f \right)$
$\quad V_t \leftarrow V$
**end**

**if** U or V not Boolean       *(i.e., if the above was aborted early (cf. Fig. 2))*
$\quad$ let $\lambda' \in \mathbb{R}$ be huge
$\quad U \leftarrow \lfloor \mathrm{prox}_{0.5, \lambda'}(U) \rceil$
$\quad V \leftarrow \lfloor \mathrm{prox}_{0.5, \lambda'}(V) \rceil$
**end**

**return** U, V

---

# C   Reproducibility

Here, we explain (**1**) how to obtain the datasets, (**2**) how we binarized them, (**3**) how to obtain the source code, and (**4**) how we parameterized the algorithms, starting with the dataset descriptions.

Table 1: Our datasets are from different domains and cover a wide-range of dimensionalties. We provide an overview over the real-world datasets involved in this study, listing their dimensionalities, densities, and selected target matrix ranks $k$ (number of components) used in our experiments.

| Dataset       | Rank | Rows   | Columns | Density |
|---------------|------|--------|---------|---------|
| Genomes       | 28   | 2 504  | 226 623 | 0.1043  |
| String        | 100  | 19 385 | 19 385  | 0.0318  |
| GBM           | 100  | 650    | 10 701  | 0.0566  |
| LGG           | 32   | 644    | 29 374  | 0.0729  |
| Cerebellum    | 450  | 644    | 30 243  | 0.0823  |
| TCGA          | 33   | 10 459 | 20 530  | 0.0501  |
| Movielens 10M | 20   | 71 567 | 65 133  | 0.0011  |
| Netflix       | 20   | 17 770 | 480 189 | 0.0067  |
| Patents       | 136  | 10 499 | 10 511  | 0.1305  |

Broadly speaking, our datasets cover three domains: the *biomedical domain*, the *entertainment domain*, and the *innovation domain*. To cover the *biomedical domain*, we extract the network containing empirical evidence of protein-protein interactions in *homo sapiens* from the STRING database. For this, we remove all interactions in the protein-protein network, for which the empirical-evidence sub-score of the STRING database is zero, retaining only the protein-protein interactions discovered experimentally. From the GRAND repository, we take the gene regulatory networks sampled from *Glioblastoma (GBM)* and *Lower Grade Glioma (LGG)* brain cancer tissues, as well as from non-cancerous *Cerebellum* tissue. These networks all stem from a method, PANDA, which extracts gene-regulatory networks from data. The weights are between $-20$ and $20$, where $-20$ corresponds to a low interaction likelihood, and $20$ corresponds to a high interaction likelihood. To binarize, we set everything to zero except the weights whose $z$-scores are in the upper $5\%$ quantile, retaining the information about likely interactions. The *TCGA* dataset contains gene expressions from cancer patients. To binarize these logarithmic expression rates $(\log_{10}(x+1))$, we again only set those weights to one whose $z$-scores lie in the upper $5\%$ quantile [23], retaining high gene expressions. We further obtain the single nucleotide polymorphism (SNP) mutation data from the *1k Genomes* project, and follow the data retrieval steps from the authors of BINAPS [9], which immediately produces a binary dataset. In the *entertainment domain*, we use the user-movie datasets *Movielens* and *Netflix*. Since we are only interested in recommending good movies, we binarize the original 5-star-scaled ratings, by setting reviews with more than 3.5 stars to one, and everything else to zero. Finally, as data from the *innovation domain*, we derive a directed citation network between patent groups from patent citation and classification data provided by *PatentsView*. We binarize this weighted network simply by setting every non-zero weight to one, retaining all edges in the network. For each of these publicly[3] available dataset, we give its dimensionality, density, and the matrix rank used in our experiments in Table 1.

In our experiments in Sec. 4, we compare ELBMF against six methods: four dedicated BMF methods (ASSO [24], GRECOND [6], ORM [35], and PRIMP [15]), one streaming Bi-Clustering algorithm SOFA [29], one elastic-net-regularized NMF method leveraging proximal gradient descent (NMF [31, 21, 22]), and one interpretable Boolean autoencoder (BINAPS [9]). The code for ASSO, GRECOND, PRIMP, SOFA, ORM, and BINAPS was written by their respective authors and is publicly available. We implement NMF and ELBMF in the Julia programming language and provide their source code for reproducibility.[4] On *TCGA*, *Genomes*, *Movielens*, *Netflix*, and *Patents*, we set the $l_2$-regularizer $\lambda = 0.001$, the $l_1$-regularizer $\kappa = 0.005$, and the regularization rate to $\nu_t = 1.0033^t$. On *GBM*, *LGG*, and *Cerebellum*, we set the $l_2$-regularizer $\lambda = 0.001$, the $l_1$-regularizer $\kappa = 0.001$, and the regularization rate to $\nu_t = 1.0015^t$. We run NMF, ELBMF, PRIMP for at most $1\,500$ epochs on each dataset. In the case that ELBMF reaches its maximum number of iterations without convergence, we bridge the remaining integrality gap simply by applying our proximal operator (see Fig. 2). To obtain a good reconstruction for PRIMP, we use a grid-width of $0.01$. To obtain a binary solution from NMF, we first clamp and then round its factor matrices upon convergence.

We set ASSO's threshold, gain for covering, and penalty for over-covering each to $1$. To achieve a better performance with ASSO, we parallelize ASSO on 16 CPU cores. Further, because ASSO's runtime scales with the number of columns, we reconstruct the *transposed* target whenever it has more columns than rows (see Table 1). For example, transposing *GBM*, *LGG*, *Cerebellum*, and *Genomes* is particularly beneficial for ASSO, as these datasets have orders-of-magnitude more columns than rows.

# D    Additional Results on the Performance of ELBMF on Synthetic Data

In this section, we provide additional results on our synthetic experiments in Sec. 4.1 and Fig. 3.

In our synthetic experiments in Sec. 4.1, we simulate data by generating *non-overlapping* tiles using rejection sampling. That is, before we place the next randomly drawn tile into our matrix, we check for overlap with past placements. If we detect an overlap, we reject, redraw, and repeat, until we placed the desired number of tiles into our matrix. In the following, we allow *overlapping* tiles, for

---

[3] GRAND.NETWORKMEDICINE.ORG      STRING-DB.ORG      CANCER.GOV/TCGA      INTERNATIONALGENOME.ORG
PATENTSVIEW.ORG GROUPLENS.ORG/DATASETS/MOVIELENS KAGGLE.COM/DATASETS/NETFLIX-INC/NETFLIX-PRIZE-DATA

[4] CS.UEF.FI/~PAULI/BASSO      GITHUB.COM/MARTIN-TRNECKA/MATRIX-FACTORIZATION-ALGORITHMS
BITBUCKET.ORG/NP84/PALTILING      CS.UEF.FI/~PAULI/BMF/SOFA      EDA.MMCI.UNI-SAARLAND.DE/PRJ/BINAPS
GITHUB.COM/TAMMOR/LOGICALFACTORISATIONMACHINES      DOI.ORG/10.5281/ZENODO.7187021

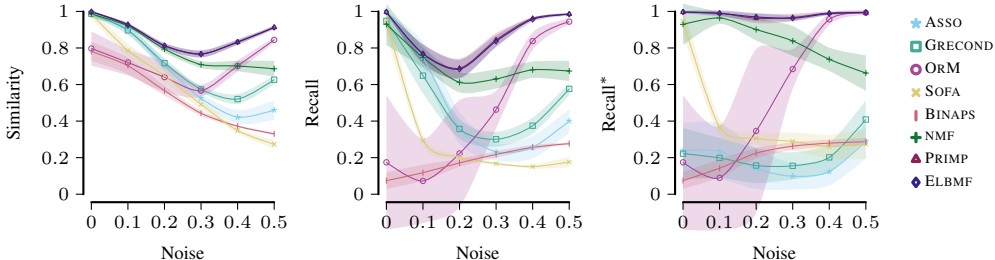

Figure 8: Overall, ELBMF reconstructs the noisy synthetic data well and recovers the ground-truth tiles which are *overlapping*. On synthetic data for additive noise levels increasing from $0\%$ to $50\%$, we show mean as line and standard deviation as shade of *similarity*, *recall* w.r.t. the target matrix, and *recall** w.r.t. the noise-free ground-truth tiles, for BINAPS, ASSO, GRECOND, OrM, SOFA, NMF, PRIMP, and ELBMF.

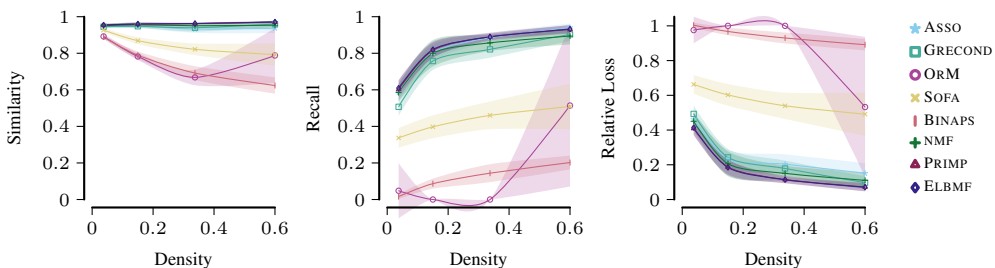

Figure 9: ELBMF reconstructs the low-noise synthetic high- and low-density matrices well and consistently so. On synthetic data with fixed additive noise level of as low as $5\%$, and increasing density, we show mean as line and standard deviation as shade of *similarity*, *recall*, and *relative loss* w.r.t. the target matrix, for BINAPS, ASSO, GRECOND, OrM, SOFA, NMF, PRIMP, and ELBMF.

which we simply omit the rejection step laid out above. To depict results on harder-to-separate data, we generate synthetic matrices as described in Sec. 4, however, this time, allowing tiles to overlap arbitrarily. In Fig. 8, we show similarity, recall, and recall*for the *overlapping case*, observing a similar behavior to Fig. 3 across the board. Again noticeable is the surprisingly good performance of rounded NMF reconstructions, outperforming ASSO, GRECOND, SOFA, and BINAPS by a large margin. Overall, PRIMP and ELBMF outperform ASSO, GRECOND, OrM, SOFA, BINAPS, and NMF across varying noise levels in similarity, recall, and recall*. Fig. 3 and Fig. 8 show that ELBMF, which does not use any post-processing, achieves best-in-class results for *overlapping* and *non-overlapping* tiles, on par with the strongest competitor, which relies heavily on post-processing.

# E  Additional Results on the Performance of ELBMF on Synthetic Data with Varying Densities

Continuing our experiments from Sec. 4.1 and Fig. 4, we ask whether our observations carry over to a low-noise scenario, in which ASSO and GRECOND performance improves significantly (see Fig. 3).

For this, we study the effects of varying densities under a low noise level of only $5\%$. As tiny tiles are hard to distinguish from noise, we see an overall improvement with increasing density, regardless of the method. With less noise, ASSO, GRECOND, and NMF improve significantly in comparison to their performance under more noise (Fig. 4). They, however, are still outperformed by PRIMP and ELBMF in recall and loss. The similarities of ASSO, GRECOND, NMF, PRIMP, and ELBMF are close to 1, whereas SOFA, BINAPS, and OrM exhibit lower similarity with increasing density. From Fig. 4 and Fig. 9, we see that ELBMF performs consistently well across varying densities, regardless of the noise level.