# OpenReview forum: "Efficiently Factorizing Boolean Matrices using Proximal Gradient Descent"
_NeurIPS.cc/2022/Conference — NeurIPS 2022 Accept_

### Official Review · Reviewer_fNGy · 2022-07-10

**Rating:** 6
**Confidence:** 4
**Soundness:** 3 good
**Presentation:** 4 excellent
**Contribution:** 3 good

**Summary:**

The authors propose a continuous relaxation to Boolean matrix factorisation that makes the problem amenable to gradient descent. A regularizer term mediates the relaxation and is consecutively reduced to obtain a Boolean solution. The method is shown to perform well on synthetic data and on-par or favorable compared to alternative methods on real-world data. The authors demonstrate that it infers semantically meaningful factors on a genomic example.

**Questions:**

- l. 6 "efficient" is meaningless without metric or comparison. Same in l 43., "elegant, accurate and efficient" without context are void terms.

- l. 24, the authors claim that methods like PCA are not interpretable for Boolean data because they have a different domain. I don't think this reasoning is self-evident. E.g., if one uses PCA with a link function with binary output, the results should be interpretable in some ways. Maybe the reasoning here could be a bit more specific?

- l. 56-58: Unnecessary jargon. BMF can be introduced in much simpler terms.

- Instead of MDL, can you assess the generalization performance on held-out data (heldout matrix entries)?

**Limitations:**

Properly addressed.

**Strengths And Weaknesses:**

- Well and clearly written, easy to follow. Thank you!

- From the empirical results (especially Fig. 5) the claim that the method is improves over SOTA is not clearly warranted. I think this is a major limitation.

- Some competitors are missing from the comparison. E.g., why no comparison with Rukat et al? (Probabilistic sampling approach - https://proceedings.mlr.press/v70/rukat17a.html).

---

> ### Author Response · Authors · 2022-08-02
> **Regarding your Review**
>
> Dear fNGy, thank you for thoroughly reviewing our submission.
> We understand and address your concerns below, starting with our choice of terminology.
>
> ## Regarding Strength and Weaknesses
>
> > From the empirical results (especially Fig. 5) the claim that the method is improves over SOTA is not clearly warranted. I think this is a major limitation.
>
> Although Hess et al.'s Primp achieves reconstruction errors which are close to ours, they only do so by relying on additional post-processing procedures, which are absent from Elbmf. Elbmf is among the best-performing methods across the board, clearly showcasing that continuous relaxations are a competitive research avenue for BMF. To be precise, our method is conceptually cleaner than the strongest-performing competitors while being consistently competitive in terms of performance. We consider this to be an improvement over the state of the art but amended the language in the manuscript to clarify the composition of this improvement.
>
> > Some competitors are missing from the comparison. E.g., why no comparison with Rukat et al? (Probabilistic sampling approach - https://proceedings.mlr.press/v70/rukat17a.html).
>
> Following your suggestion, we now additionally compare with Rukat et al.'s OrM in all experiments of our updated paper, thus also covering probabilistic sampling approaches to Boolean Matrix Factorization. In our experiments, we outperform OrM in recall, similarity, and loss, on both real-world and synthetic data.
>
> ## Regarding Questions
>
> > l. 6 "efficient" is meaningless without metric or comparison. Same in l 43., "elegant, accurate and efficient" without context are void terms.
>
> We appreciate your general sensitivity to un(der)defined adjectives, which we used in the introduction to concisely summarize what we perceive as the core strengths of our method, hoping that their precise meaning would become clear in the remainder of the paper.
> In our usage, efficient served to underline scalability (e.g., to large real-world datasets), elegant served to emphasize conceptual simplicity (e.g., no post-processing needed), and accurate served to highlight result quality (i.e., good reconstructions).
> We have amended the writing in the affected passages of our manuscript for clarification by substituting the adjectives in question with their implied meaning.
>
> > l. 24, the authors claim that methods like PCA are not interpretable for Boolean data because they have a different domain. I don't think this reasoning is self-evident. E.g., if one uses PCA with a link function with binary output, the results should be interpretable in some ways. Maybe the reasoning here could be a bit more specific?
>
> We used the term interpretability in the sense of direct interpretability, i.e., the outputs of the method can be interpreted by a domain expert using domain knowledge without any post-processing. The outputs of methods like PCA might be made interpretable by post-processing, but this additional indirection increases the risk of misinterpretation.
> We amended the language in the manuscript to clarify our reasoning.
>
> > l. 56-58: Unnecessary jargon. BMF can be introduced in much simpler terms.
>
> We have simplified the introduction of BMF to avoid the jargon you identified.
>
> > Instead of MDL, can you assess the generalization performance on held-out data (heldout matrix entries)?
>
> We believe that this should be possible both in theory and in practice. There is very good theory on how to assess the generalization performance of NMF or SVD using bi-cross-validation [1] by randomly selecting held-out matrix entries to assess generalization performance. As far as we know, there is, however, no specialization for BMF. In fact, we started out with two approaches from Owen and Perry [1], but both were practically unable to computationally scale to the size of our experiments. We do, however, consider scalable bi-cross-validation for BMF to be an interesting avenue for future work.
>
> [1] Owen and Perry. "Bi-cross-validation of the SVD and the nonnegative matrix factorization." The annals of applied statistics 3.2 (2009): 564-594.
>
> Thank you for the critical feedback on our submission, which helped us further improve our manuscript. If we have answered your questions to your satisfaction, and should you feel that the corresponding changes to our manuscript further raised the quality of our submission, we very much hope that you consider reflecting this in your score.

---

### Official Review · Reviewer_t3GQ · 2022-07-11

**Rating:** 4
**Confidence:** 3
**Soundness:** 2 fair
**Presentation:** 3 good
**Contribution:** 2 fair

**Summary:**

This paper relaxed Boolean matrix factorization under linear algebra. They proposed a novel elastic BMF regularizer to ensure Boolean factors for the decomposition. They also showed useful applications of the proposed methods in the analysis of biological data.

**Questions:**

As in weaknesses, my main concerns as the advantage of relaxing compared with traditional BMF and the experimental setting.

**Ethics Review Area:**

["I don’t know"]

**Strengths And Weaknesses:**

Strengths:
1. The proposed elastic BMF regularizer that penalize the value distribution to ensure the Boolean factors.
2. The proposed method showed empirical better performance with some classic BMF methods.
3. They also showed promising applications across different domains.

Weaknesses:
1. Binary matrix can easily be high rank under linear algebra, thus prevent the application of many classic methods like SVD. While under Boolean algebra, optimally, the rank of the matrix can be reduced to its log level. As far as I regard, this is the main advantage to apply BMF in analyzing binary data. By relaxing the problem to NMF under linear algebra while ensuring Boolean factors does not solve the high rank issue of binary matrix. It thus posed a question of how practical this proposed method is useful in analyzing binary data.
2. The experimental design is not quite fair. The data is synthesized with rectangular non-overlapping tiles, which is quite rear for binary data, where overlapping is very common.
3. I am also confused that in figure 2b, why the error converged not on the smallest error. Is it because the penalization of Boolean factors cost extra error in the factorization? If so, how to quantify such tradeoff?
4. The author discussed lots of related methods but only compared with very few of them. Like the most recently probabilistic methods from Ravanbakhsh et al, Rukat et al. As for the relaxing method, I also notice a rather recent method called Faststep: Scalable boolean matrix decomposition.

---

> ### Author Response · Authors · 2022-08-02
> **Regarding Weaknesses**
>
>
> > [...] It thus posed a question of how practical this proposed method is useful in analyzing binary data.
> [Unfortunately, we had to abbreviate your question due to the word limit.]
>
> We fully agree with your assessment that it is highly problematic to apply standard Matrix Factorization techniques that use continuous linear algebra, such as SVD, to Boolean matrices. This is reflected in lines 23–45 of our manuscript, where we argue that we can benefit from efficient gradient-based continuous optimization techniques also in Boolean Matrix Factorization if appropriately regularized or even rounded. Due to our regularization, we achieve very sparse and low-rank representations upon convergence. We also acknowledge in Section 5, Paragraph 2 ('Limitations') that in the early phase of our algorithm, the intermediate factor matrices are not sparse and essentially result in a full-rank reconstruction, but this hardly affects the overall utility of our method. Experimentally, we see that our method is as sparse as, or even sparser than, the BMF methods based on combinatorial algorithms, such as Asso and GreConD, i.e., Elbmf results in low-rank reconstructions (Fig. 8). In our experiments, we see that methods based on continuous relaxations are competitive in this regard, and the experiments from Hess et al., Zhang et al., and Araujo et al. (FastStep) also all show that continuous relaxations can result in low-rank reconstructions of Boolean matrices using linear algebra.
>
> > The experimental design is not quite fair. The data is synthesized with rectangular non-overlapping tiles, which is quite rear for binary data, where overlapping is very common.
>
> Our experimental setup is fair in the sense that we use the same datasets for all competing methods, but we recognize your concern that our experimental design may not be sufficiently realistic. In our original setup, the experiments on synthetic data were designed to create a setting that would allow us to unambiguously assess the quality of our reconstruction vs. the ground truth, and we saved realism for the experiments on real-world data. In response to your critique, however, we now include an additional experiment on synthetic data generated with overlapping tiles in the Appendix C, Figure 7.
>
> > I am also confused that in figure 2b, why the error converged not on the smallest error. Is it because the penalization of Boolean factors cost extra error in the factorization? If so, how to quantify such tradeoff?
>
> In Fig. 2b, we observe an effect caused by the problem that you raised in point 1. That is, we see that a less constrained factorization results in a reconstruction causing lower loss than a highly constrained factorization. In other words, it is much cheaper for, e.g., an SVD to result in a continuous, high-rank reconstruction. This is exactly what prohibits the usage of methods using continuous linear algebra on Boolean data---unless they are properly regularized like Elbmf.
> Regarding the quantification of the trade-off: Thank you for this interesting question. In short: We currently don't know. In the context of Integer Linear Programming (ILP), there exists a technique called randomized rounding that can provide approximation guarantees when applied to a solution of a continuously relaxed problem. As BMF is, unfortunately, a non-linear problem, we believe that the trade-off quantification in our case is not as easy. However, this sounds like a good starting point for further research into continuous relaxations of BMF.
>
> > The author discussed lots of related methods but only compared with very few of them. Like the most recently probabilistic methods from Ravanbakhsh et al, Rukat et al. As for the relaxing method, I also notice a rather recent method called Faststep: Scalable boolean matrix decomposition.
>
> Thank you for bringing Araujo et al.'s FastStep to our attention, which we were not previously aware of. FastStep also considers continuous relaxations, gradient based optimization using linear algebra, to return Boolean matrices efficiently at scale, not unlike Hess's Primp, Zhang's BMF, or our Elbmf. As this work is clearly related, we updated our related work section accordingly.
> Following your suggestion, we now also compare with OrM from Rukat et al. as part of our updated experiments section, thus also covering the state of the art in probabilistic Boolean Matrix Factorization. In our experiments, we outperform OrM in recall, similarity, and loss; both on real world and synthetic data.

---

> ### Author Response · Authors · 2022-08-02
> **Regarding Questions**
>
> > As in weaknesses, my main concerns as the advantage of relaxing compared with traditional BMF and the experimental setting.
>
> We addressed your concerns regarding missing competitors by updating our experiments, and your concerns regarding continuous relaxations of the BMF problem in the discussion above.
>
> Thank you for your feedback, including the valuable suggestions of FastStep as related work and Rukat et al.'s OrM as an additional competitor, which helped us improve our work. We hope that following our explanations above and the accompanying changes to our manuscript, you now have more confidence that continuous relaxations are a valid approach for BMF. If you are satisfied with our answers and the corresponding changes to our submission, we sincerely hope that you consider reflecting this in your score.

---

> > ### Comment · Reviewer_t3GQ · 2022-08-08
> > **Thanks for the response.**
> >
> > Thanks for the response. But I am still not fully persuaded by the results.
> >
> > For the synthetic data, the experimental setting is very simple. It only considered one tile situation but with a different level of noise. Such comparison alone does not illustrate the efficiency of the methods. And the added comparison by "allowing overlapping" is a very vague description too. An easy way to workaround this is to follow the simulation methods of recent work like in Wan et al (AAAI 2020) by considering different density levels and noise levels. In the new result, the OrM does not show numerical stability, I also mentioned the method from Ravanbakhsh et al (ICML2016). Per my experience, this method is more stable than OrM. I would also like to see the result in comparison.

---

> > > ### Author Response · Authors · 2022-08-08
> > > **Synthetic data and additional competitors**
> > >
> > > We thank you for your feedback and are happy that you are satisfied with the first part of our response. In the following, we address the remaining points.
> > >
> > > ## Regarding our synthetic experiments
> > >
> > > ### Number of tiles
> > >
> > > We agree that considering one tile is insufficient to realistically assess performance, which is exactly why we already consider scenarios with more than one tile in our experiments, as detailed in the following. We have clarified the wording in the manuscript to ensure this important point does not get lost (lines 214, 222, and 224).
> > >
> > > For either overlapping or non-overlapping tiles, we generate 10 matrices (Experiment 4.2, Appendix C), each containing 5 randomly drawn tiles, each of randomly selected width and height (to account for different densities), for each of our 6 noise levels. In other words, we consider 100 tiles with different densities for each of the 6 noise levels.
> > >
> > > ### Varying densities
> > >
> > > We agree with the assessment that varying densities and noise levels is interesting in the context of synthetic experiments, which is exactly why we vary both in our Experiment 4.2, as detailed in the following. Again, we have clarified the wording in the manuscript to ensure this important point does not get lost (line 192).
> > >
> > > In Experiment 4.2 (Appendix C), we started out with non-overlapping tiles in the first place, to make it simple to control the density range precisely.
> > > To ensure different densities, we randomly draw width and height of each tile independently, thus ultimately obtaining matrices of different densities in the range of 0.1 to 0.4 (before noise). Starting with differently dense ground-truth tiles, we then add different levels of noise, thus varying the chance of setting matrix cells to $1$, and further varying the overall density.
> > >
> > > ### Overlapping
> > >
> > > We elaborate on the meaning of "allowing overlapping" below, and have amended the manuscript to clarify this point (line 555).
> > >
> > > In the experiments without tile overlap, we perform rejection sampling.
> > > That is, before we place the next randomly drawn tile into our matrix, we check for overlap with past placements. If we detect an overlap, we reject, redraw, and repeat, until we placed the desired number of tiles into our matrix.
> > > In the experiments allowing overlapping tiles, we do not check for overlap, thus inserting each draw into the matrix.
> > >
> > > ## Regarding Rukat’s OrM and additional competitors
> > >
> > > We manually optimized hyper-parameters for OrM to arrive at the best result, which we report in the submission. Encouraged by your response, we started preliminary experiments with Ravanbakhsh et al.’s MP, but it is unlikely that MP will finish before the discussion period ends. We will happily include Ravanbakhsh et al.’s MP method as a competitor in the camera-ready version. This will grow the number of competitors to 8, i.e., 2-3 times the number of competitors seen in prior work on BMF, and should convince even the toughest critics that our method improves over the state of the art.
> > >
> > > We hope that our additional elaboration and the corresponding changes to our manuscript, along with the prospective inclusion of MP as an additional competitor in the in-camera version of the paper, have dispelled your remaining doubts regarding our method and our results. In this case, we would very much appreciate if you reflected this in your score.

---

> > > > ### Comment · Reviewer_t3GQ · 2022-08-08
> > > > **Appreciate the response**
> > > >
> > > > Appreciate the response. But it would also be difficult for me to increase the score based on the future inclusion of experiments in the camera ready version since I have mentioned both methods in my very initial comments. Another question I have is that as there are different density level, will the performance be different? Does the proposed method did well in both dense and sparse scenarios.

---

> > > > > ### Author Response · Authors · 2022-08-08
> > > > > **Performance under varying densities**
> > > > >
> > > > > In response to your initial comments, we already planned the inclusion of both Ravanbakhsh et al.’s method and Rukat et al.’s method. As Rukat was mentioned twice and Ravanbakhsh once, we naturally started with the former. This left us with little time to complete the inclusion of Ravanbakhsh’s method. We are confident that our method will outperform this method just like it did its other 7 competitors, but we acknowledge that you might not be convinced without seeing the final results. In this case, we would still appreciate if you considered reflecting in your score that we addressed all of the other points you raised within the discussion period (and will be addressing the last one in the final version).
> > > > >
> > > > > In our experiments, we see no scenario in which Elbmf clearly falls short, including particularly high or low densities. Our real-world data cover very sparse matrices, such as the Movielens 10M (0.11%), on which we perform just as well as on much denser datasets, such as Patents (13%) or Genomes (10%). In our synthetic experiments, we observe that we robustly reconstruct the ground-truth tiles even as noise levels (and, consequently, densities) increase, whereas most other methods tend to perform increasingly worse due to the noise.

---

> > > > > > ### Comment · Reviewer_t3GQ · 2022-08-08
> > > > > > **Why I take very serious consideration on experiments**
> > > > > >
> > > > > > The advantage of BMF over NMF or the linear relaxation has been numerically proved. Under Boolean algebra, the matrix rank for binary matrix can be reduced to the log level of the matrix rank in linear algebra. What this paper proposed here partly contradicts such proof, which essentially raised the requirement for the experiments. That is the reason I question the experiment process and ask for more details about different data scenarios like different density levels, noise levels, and baseline methods with different perspectives. I appreciate the author's response. However, I still feel the rebuttal does not give me enough confidence. Specifically, it would not be a favorite excuse to go for only one of the two methods as I mentioned one twice or once. Also, talking about experiments, figures, tables, and data points would be more favorable than descriptive words like "we see no scenario in which Elbmf clearly falls short".

---

> > > > > > > ### Author Response · Authors · 2022-08-09
> > > > > > > **Regarding Boolean Rank**
> > > > > > >
> > > > > > > # Response
> > > > > > >
> > > > > > > Thank you again for your attention to detail regarding our experiments.
> > > > > > > We hope that from our prior exchange, it is clear that our experiments tested Elbmf and its competitors in many different data scenarios, including different density levels and noise levels, and that we are including (and were working to include additional) baselines with different perspectives. In an additional attempt to convince you that our method improves over the state of the art, we address the points you raised in your latest comment below.
> > > > > > >
> > > > > > > ## Boolean Rank
> > > > > > >
> > > > > > > You are right in that extraordinary claims require extraordinary evidence.
> > > > > > > We, however, do not claim to discover a result that contradicts the theorem regarding the Boolean matrix rank. The reason is that this theorem is after the _exact_ Boolean matrix rank, whereas we are after approximate and inexact low-rank reconstructions (line 26-27) or inexact and approximate matrix ranks (cf. line 238 and Figure 4) that do not overfit the matrix or the noise within it. This is why it is possible to use relaxations to estimate an inexact but useful matrix rank in the first place. Furthermore, it is common practice in the BMF community to deploy model selection criteria (BIC, AIC, MDL, Bai & Ng’s, or Bi-Cross-Validation) to prevent the overfitting.
> > > > > > >
> > > > > > > Preventing overfitting when determining the inexact Boolean rank is exactly the problem we study experimentally in Experiment 4.1. There, we are explicitly not after the exact Boolean matrix rank, but rather after an inexact approximation of the simulated matrix rank, which only corresponds to the ground-truth tiles and not to the noisy bits (line 238).
> > > > > > > This experiment shows us that we can select a non-overfitting inexact reconstruction rank. Put simply, our discovered rank is _not_ the exact Boolean rank.
> > > > > > >
> > > > > > > By accident, we changed the phrasing from ground-truth Boolean rank to true Boolean rank for brevity, which might be the cause of this misunderstanding. To clarify that we are after an inexact matrix rank and are not contradicting a general theorem, we amended our phrasing in our abstract (line 233) and our experiments (line 233-235, line 10).

---

> > > > > > > ### Author Response · Authors · 2022-08-09
> > > > > > > **Concerning your specific points**
> > > > > > >
> > > > > > >
> > > > > > > ## Concerning your specific points
> > > > > > >
> > > > > > > To clarify, we prioritized the experiments with OrM over those with Ravanbakhsh et al.’s method because OrM was mentioned by one of the other reviewers as well (this is what we meant by "mentioned twice"); we did not state this as an "excuse" but rather to be transparent about our scheduling decisions. The experiments with Ravanbakhsh et al.’s method, which necessitated a separate python2 (!) setup, were slowed down by the fact that the original code available online was buggy, but we managed to resolve this issue. Running Ravanbakhsh et al.’s method, we noticed that its memory requirements are very large, prohibiting us from allocating enough memory on a cluster for almost all real-world datasets used in our experiments: TCGA, Movielens 10M, Netflix, Patents, PPI, StringDB, Cerebellum, Genomes. In contrast, Ravanbakhsh et al.’s 2016 considers only the much smaller Movielens 1M, Movielens 100K, or Senate Voting Records matrices (with rows and columns in the order of hundreds). As we are after scalability, we consider much larger matrices with much higher row and column counts (Table 1, Appendix B), thus requiring more resources. As a potential resolution going forward, we will reimplement this method in a more efficient programming language. This, however, requires more time to ensure correctness.
> > > > > > >
> > > > > > > Again to clarify, our "descriptive words" served to summarize our overall findings, namely, that Elbmf performed well across the board in our experiments, including under different density and noise levels. We now include an __additional experiment showing performance at varying density levels__ for densities from 0.0375 to 0.6, before additive noise (0.05), __this time also including Ravanbakhsh’s method (MP for Message Passing)__, based on its tutorial and demo code. In the results, which we report in Fig. 8 (Appendix E), we can see that the increasing density affects the performance of all methods, however, it does not affect the performance of all methods equally.  Asso, GreCond, NMF, Sofa, Binaps, OrM, Primp, and Elbmf all improve in recall and loss with increasing density. This is because very tiny tiles in the beginning are very hard to distinguish from noise. The methods Asso, GreCond, and NMF are again competitive, however, still easily outperformed by Primp and Elbmf. Closely trailing Binaps, MP is consistently in the last place, unable to recover the ground-truth tiles, regardless of the density level. With increasing density, OrM gets slightly worse before its performance increases dramatically, such that it finishes on a recall, similarity, and loss similar to Sofa.
> > > > > > >
> > > > > > > The similarity, Fig. 8a, however, is consistently close to 1 for Asso, GreCond, NMF, Primp, and Elbmf, but drops with increasing density for Sofa, Binaps, OrM, and MP. The reason is that a 0-heavy reconstruction is getting progressively worse for increasingly dense matrices. This new experiment now shows clearly that Elbmf is the top-perforrming method across all density levels, outperforming 8 competitors, including Ravanbakhsh et al.’s method, MP.
> > > > > > >
> > > > > > > We sincerely hope that our additional explanations and the changes to our submission managed to increase your confidence in our results, perhaps to the point that you might consider reflecting this in your score.

---

> > > > > > > > ### Comment · Reviewer_t3GQ · 2022-08-09
> > > > > > > > **No code provided**
> > > > > > > >
> > > > > > > > It seems like the dropbox link provided in the paper only contains a copy of the manuscript. Does not include any code of algorithm and the experiments.

---

> > > > > > > > > ### Author Response · Authors · 2022-08-09
> > > > > > > > > **We provide Code as Supplementary Material in OpenReview, now also via Dropbox**
> > > > > > > > >
> > > > > > > > > We uploaded the code as supplementary material to OpenReview when submitting our first revision.
> > > > > > > > > To improve discoverability, we now also uploaded it to Dropbox.
> > > > > > > > >
> > > > > > > > > We included the DropBox link as a placeholder to anonymize our submission.
> > > > > > > > > In the camera ready version, we want to replace the DropBox link with a link to a proper GitHub page, including our code to reproduce our experiments and scripts to fully automate the process.

---

### Official Review · Reviewer_Bmfn · 2022-07-12

**Rating:** 7
**Confidence:** 4
**Soundness:** 3 good
**Presentation:** 3 good
**Contribution:** 3 good

**Summary:**

The authors propose an elastic net inspired approach for the optimization of Boolean Matrix Factorizations (BMF). Relying on a relaxed objective with regularization towards binary values, they derive the proximal operator for the proposed regularization, enabling the use of Proximal Alternating Minimization. A gradual increase of the regularizer weights is supposed to alleviate the problem of being trapped in local minimizers. The experimental analysis shows that the proposed method maintains desirable properties, such as robustness to noise and feasibility to determine the rank of the factorization, while being less computationally expensive.

**Questions:**

* How does the gradual increase of the regularization weights work? Do you not optimize the binary MF objective with this approach?
* How much overlap between clusters (tiles) is in the synthetic data?

# After Rebuttal Thoughts
As I laid out in my author's response, I think that there is an obvious contribution of the proposed elastic net binary penalizer. I also think that the optimization with a gradual increase of the penalization weights leads to a good optimization scheme that is less likely to be stuck in unsuitable local minimizers. I think that the framing of the proposed method is maybe not optimal, but it's not exactly wrong. Because I think that the authors will be able to improve the framing and because I see the contribution of the paper, I increase my score to accept.

**Limitations:**

There is no designated Limitations section but a broader impact section, describing potential ethical aspects. I think this should be clear, that methods as the proposed one can be possibly misused. If it were according to me, I would say the authors can scrap this section and instead explain the method a bit more in detail.
Apart from the mentioned question about binary vs. Boolean factorizations, I don't see any limitations that should be discussed here.

**Strengths And Weaknesses:**

# Strengths
* The paper is well-written and good to follow (by and large)
* The proposed approach is extremely sound (except for one thing mentioned below). Applying best practices from L1- and L2- regularization to the L1- and L2- inspired binary regularization terms makes a lot of sense. I also expect that the gradual increase of the weights helps to obtain better minimizers (although this doesn't show that much in the experiments)
* The experimental analysis is well done, addressing many relevant aspects of BMF (determination of the rank, robustness, synthetic and real-world data)

# Weaknesses
* I take from the paper that the matrices are optimized by PALM until they are binary. This is weird because that would mean that the factorization is computed in elementary algebra. A strongly  regularized elastic BMF is equivalent to solving the objective of binary matrix factorization:
 $$\min \lVert A-UV\rVert^2 \text{ s.t. }U\in\\{0,1\\}^{n\times k}, V\in\\{0,1\\}^{k\times m}$$
However, that is not the objective of Boolean factorization, where the matrix product is computed in Boolean algebra. That is why Primp relies on a post-processing step: a relaxed, almost binary result can mock the properties of the BMF after rounding.

I would be happy to accept this paper, but the issue above should be clarified before I can vote for acceptance.

# Minor Issues/Comments
* What is the definition of $\oplus$ in Eq. (2)?
* How the gradual increase of the regularization weights is performed, is a bit unclear. The text states that in every iteration the weight is increased by multiplication with $\nu_t$. However, I didn't see in the experiments section how $\nu_t$ is determined.
* The MDL choice is not very well motivated. I see that there might be space restrictions, but maybe this could be a bit more explained.
* Some of the terminologies are not really introduced, like tiles.

---

> ### Author Response · Authors · 2022-08-02
> **Regarding Strength and Weaknesses**
>
> Dear Bmfn, thank you for thoroughly reviewing our paper. We appreciate your concerns and will address each one in the following, starting with your question regarding the difference between binary and Boolean matrix factorization.
>
> Your concern regarding binary and Boolean matrix factorization might be a terminological misunderstanding. To clarify, we deliberately talk about Boolean matrices, rather than binary matrices, as the former are almost universally recognized as $\{0,1\}$-matrices, whereas the latter are sometimes used to denote $\{-1,+1\}$-matrices (e.g., in the Quantized Learning of DNNs Community). Although based on efficient continuous linear algebra, our method still shares its core goals with the Boolean Matrix Factorization Community, thus we believe that our naming convention actually reflects this well. We are, however, open to changing our naming convention in case this helps prevent confusion.
>
> In a nutshell, the problem addressed by both Primp and Elbmf is that binarizing the relaxed BMF solution using simple rounding toward the nearest feasible solution can significantly worsen the objective function. Primp prevents this with a quadratic thresholding procedure as a post-processing step, whereas Elbmf uses a regularization scheme which results in Boolean solutions upon convergence, or in an almost-Boolean solution that can be binarized without worsening the loss significantly. In experiment 4.1, we asked whether binarizing projections increases the objective function significantly, and we provide the answer for using either a constant regularization (e.g., Primp) or a monotonically increasing regularization rate (Elbmf). In Fig. 2a, we see that the Boolean gap decreases very slowly under constant regularization, which necessitates Primp's post-processing and thresholding. In contrast, the _Boolean gap swiftly approaches $0$_ under monotonically increasing regularization rate, allowing us to _safely project the solution_ (after a limited number of iterations) to a truly Boolean factorization _without causing a significant degradation_ of the objective function.
>
> In theory, applying this regularization scheme to Primp is possible. As Primp’s regularizer is $l_1$-inspired, however, changing its regularization strength in practice has a much stronger effect on the corresponding proximal operator (in comparison to Elbmf's proximal operator). Therefore, amending Primp's regularization in this way would require extra precise and careful tuning, which is not necessary when using Elbmf.
> Our factor matrices $U \in \{0,1\}^{n,k}$ and $V \in \{0,1\}^{k,m}$ are, like Primp's, truly binary. This allows us to use Boolean algebra to create a Boolean reconstruction $U \circ V$ (Eq. (1)) of the Boolean target matrix, and Boolean algebra helps us define our recall (line 212) and the rank selection criterion (line 239).
>
> We updated Sec. 2.2 to clarify how we arrive at our Boolean factor matrices, what role Boolean algebra plays in our method and in our evaluation in Sec. 4, and included this dedicated step in the long version of our pseudocode in Appendix E.

---

> > ### Comment · Reviewer_Bmfn · 2022-08-08
> > **Boolean vs Binary MF**
> >
> > Dear authors, thanks for your comments. So, if I understand correctly according to what is stated in Sec. 2.2, the idea is to push the matrices onto binary ones such that in the end the grid search is not necessary anymore and you can round at 0.5 (as it would be done by the prox operator)? I'm still a bit doubtful that this approach really fits the Boolean product. I think that your major contribution is the proposal of the prox operator for the Elastic Net version and it's good to evaluate the potential of this regularization to get rid of the regularization weight by gradually increasing it. I think this is a valid contribution and whether you apply this scheme now to Boolean factorization or to binary MF or biclustering like in [1] is left to the user's requirements and data. Maybe just write a small section that motivates the focus on Boolean MF in this paper although the proposed regularization is maybe even more naturally applicable to binary MF and biclustering. But Boolean MF is maybe the most difficult to optimize because it's in another algebra, and hence it makes sense to focus on this factorization experimentally 🤔
> >
> > [1] Hess, Sibylle, et al. "BROCCOLI: overlapping and outlier-robust biclustering through proximal stochastic gradient descent." Data Mining and Knowledge Discovery 35.6 (2021)

---

> > > ### Author Response · Authors · 2022-08-08
> > > **Boolean vs Binary MF**
> > >
> > > First of all, thank you for reflecting your increased confidence in our method in your score. In the following, we address the questions you raised in your reply to our first response, hoping to remove any remaining uncertainties.
> > >
> > > Yes, your understanding of the referenced passage in Sec. 2.2 is correct. Our regularization rate, combined with our proximal operator, makes the grid search obsolete. However, although not necessary, one can still use grid search in conjunction with our method.
> > >
> > > Rather than grid search, we use a simple-to-evaluate ‘thresholding’ in the presence of early-stopping criteria. Rounding at 0.5 is a very valid and efficient option – however, it hides the fact that we are very certain about the rounding threshold. That is, because we swiftly reach a Boolean gap of almost 0, we are rounding at a stage with vanishingly low uncertainty about the rounding direction. For example, we often saw a difference between individual matrix cells and their closest integer value (0 or 1) that is much lower than the Float64 machine epsilon (in the order of 1e-16).
> > >
> > > We evaluate using Boolean algebra but optimize using regular linear algebra for efficiency. In other words, Boolean algebra is not part of the algorithm, but only used for assessing and comparing results from all methods included in our experiments. We clarified the introduction, abstract, and theory section to underline that we use continuous relaxations while optimizing.
> > >
> > > You are again right in your assessment that it is quite natural to apply our regularizer in combination with our increasing regularization rate to bi-clustering, binary MF, or even binary autoencoders. We believe that our regularizer has a broader applicability, which we clarified in our introduction. We are very motivated to study this regularizer and the increasing regularization rate in the context of binary MF, bi-clustering, or even binary-weight autoencoders in the future.
> > >
> > > To emphasize the wide applicability of our regularizer, we renamed it to the elastic binary regularizer  (Elb). In the introduction, we note that Elb and its rate are not confined to BMF, but rather are widely applicable to regularize problems such as bi-clustering.
> > >
> > > Finally, thank you for bringing up BROCCOLI, which is clearly related to our work. We updated our related work section and our introduction to include it.

---

> ### Author Response · Authors · 2022-08-02
> **Regarding Minor Points**
>
> > What is the definition of $\oplus$ in Eq. (2)?
>
> We used $A \oplus B$ to denote the coordinate-wise logical exclusive-or (xor) between $A \in \{0,1\}^{n,m}$ and $B\in \{0,1\}^{n,m}$. We now introduce the symbol in line 62 in our revised document.
>
> > How the gradual increase of the regularization weights is performed, is a bit unclear. The text states that in every iteration the weight is increased by multiplication with $\nu_t$. However, I didn't see in the experiments section how $\nu_t$ is determined.
>
> In general, we require the choice of a scaling function $\nu_t$ (which ideally is monotonically increasing). In particular, in experiment 4.2 (Fig. 2), we choose $\lambda_t = \nu_t = 1.05^t$ (see the caption of Fig. 2). Due to the page limit, we explain all hyper-parameters used in the remainder of our experimental section in Appendix B (``Reproducibility''). To improve the discoverability of this information, we now reference Appendix B in our experiments section (line 172, footnote).
>
> > The MDL choice is not very well motivated. I see that there might be space restrictions, but maybe this could be a bit more explained.
>
> As far as we know, there is no strictly superior rank selection algorithm for Boolean Matrix Factorization that is guaranteed to perform best. Motivated by its widespread usage in related work [1,2,3,4,5,6], and its practical performance in preliminary experiments, we believe that, for now, Pauli Miettinen's MDL4BMF [4] is a robust baseline to select the Boolean matrix rank. We also experimented with bi-cross-validation specialized for SVD and NMF [7], which unfortunately was not scalable to our large real-world datasets. Since we do not exclusively focus on the interesting rank selection problem, we only briefly touched upon this topic in our experiments. Apart from our model selection experiment, which includes AIC, MDL and Bai & Ng's IC, we only select the rank for the GBM, LGG, String, and Cerebellum datasets (line 259). Model selection, thus, constitutes only a small part of our real-world experiments.
>
> We revised the motivation for the model selection to reflect that we choose a commonly used baseline technique.
>
> - [1] Hess, S., Morik, K., Piatkowski, N., 2017. The PRIMPING routine—Tiling through proximal alternating linearized minimization. Data Min Knowl Disc 31, 1090–1131.
> - [2] Hess, S., Morik, K., 2017. C-SALT: Mining Class-Specific ALTerations in Boolean Matrix Factorization, in: Machine Learning and Knowledge Discovery in Databases. Springer, pp. 547–563.
> - [3] Fischer, J., Vreeken, J., 2021. Differentiable Pattern Set Mining, in: KDD ’21, pp. 383–392.
> - [4] Miettinen, P., Vreeken, J., 2014. MDL4BMF: Minimum Description Length for Boolean Matrix Factorization. TKDD, 18:1-18:31.
> - [5] Rukat, T., Holmes, C.C., Titsias, M.K., Yau, C., 2017. Bayesian Boolean Matrix Factorisation, in: ICML, pp. 2969–2978.
> - [6] Araujo, M., Ribeiro, P., Faloutsos, C., 2016. FastStep: Scalable Boolean Matrix Decomposition, in: Advances in Knowledge Discovery and Data Mining, pp. 461–473.
> - [7] Owen and Perry. "Bi-cross-validation of the SVD and the nonnegative matrix factorization." The annals of applied statistics 3.2 (2009): 564-594.
>
> > Some of the terminologies are not really introduced, like tiles.
>
> For us, a tile is a rectangular consecutive area (block) in a matrix that is filled with ones. We clarified the notion of a tile in line 188 of the updated document.

---

> ### Author Response · Authors · 2022-08-02
> **Regarding your Questions and Limitations**
>
>
> > How does the gradual increase of the regularization weights work? Do you not optimize the binary MF objective with this approach?
>
> To scale the $l_2$-regularization weight $\lambda$, Elbmf requires an (ideally monotonically increasing) scaling function $\nu_t$ such that $\lambda_t = \lambda \cdot \nu_t$. Since we wanted to regularize weakly in the beginning (to land in a minimum) and force a Boolean solution in the end, we chose a monotonically increasing exponential form $\nu_t = (1+c)^t$ ($c \geq 0$, `small'), for example $\nu_t = 1.005^t$, but alternative scaling functions are also possible. We provide the exact hyper-parameters (such as regularization rate $\nu_t$) used in all our experiments in _Appendix B_ (``Reproducibility'').
> To aid discoverability of Appendix B, we now include a reference in Sec. 4.
>
> > Do you not optimize the binary MF objective with this approach?
>
> Yes, we make use of our scaled regularization coefficients in our objective function. More precisely, we use the scaled coefficient $\lambda_t = \lambda \cdot \nu_t$ to parameterize the regularization of our objective function $\ell_\kappa,\lambda_t$, thus immediately affecting our proximal operator. Intuitively speaking, per iteration $t = 1,2,\dots$ we increase ($\nu_t$) the projection ``distance'' of our proximal operator, starting from only small nudges, to eventually reach large binarizing leaps.
> To clarify the relationship between our scaling $\nu_t$, the objective function $\ell_{\kappa,\lambda\nu_t}$, and the proximal operator $\text{prox}_{\kappa,\lambda\nu_t}$, we made the notation explicit (Eq. (9)) and updated the submission accordingly.
> To further supplement our answers, we now include the complete iPALM-based Elbmf algorithm as pseudocode in Appendix E of our revised document.
>
> > How much overlap between clusters (tiles) is in the synthetic data?
>
> There is no overlap. To simplify the judgment of the reconstructions, we generated non-overlapping tiles in our synthetic experiments in Sec. 4 (cf. line 187).
> Motivated by your inquiry, we now additionally include experiments with overlapping tiles in Appendix C, Figure 7. There, we see almost identical performance of each method across the board.
>
> ## Regarding Limitations
>
> > There is no designated Limitations section but a broader impact section, describing potential ethical aspects. I think this should be clear, that methods as the proposed one can be possibly misused. If it were according to me, I would say the authors can scrap this section and instead explain the method a bit more in detail. Apart from the mentioned question about binary vs. Boolean factorizations, I don't see any limitations that should be discussed here.
>
> Although we included a discussion of our limitations, we had not designated the section as such. To enhance its discoverability, we endowed this part of Section 5 (Conclusion) with a separate heading. We agree that the broader impact of our method, Elbmf, is, generally speaking, clear to our community. Nevertheless, we incorporated your feedback without scrapping the broader impact section in an attempt to follow NeurIPS's best practice guide. We remain open to moving this section into the appendix, however, should this become necessary to stay within the page limit of the main paper.
>
> Thank very much for your valuable feedback and genuine interest in our submission, which clearly helped us improve both our work and its presentation. Should you be satisfied with our answers and the corresponding changes to our submission, we very much hope that you consider reflecting this in your score.

---

### Author Response · Authors · 2022-08-02
**Regarding our Revision in General**

We would like to thank all reviewers for their constructive feedback. Responding to all of their criticisms, we could improve our submission considerably. Before replying to each reviewer individually, we summarize the general points addressed in our revision:

- To address the reviewer Bmfn's questions regarding Boolean-ness, we include the necessary details in Sec. 2.2 and provide an extended version of our algorithm in Appendix E.
- To clarify the relationship between regularization rate and proximal operator, we explicitly include coefficients in our notation, reference the formal derivative of our operator in Appendix A, and give a long version of our algorithm in Appendix E.
- To increase the discoverability of our reproducibility information (such as the choice of regularization rate), we include a reference to Appendix B.
- To simplify our presentation, we contextualize our nomenclature, explained tiles, improve the argument against SVD for Boolean matrices, motivate the rank selection criteria, define the exclusive-or $\oplus$, and remove unnecessary jargon.
- To account for Araujo et al.'s FastStep, we updated the related work section.
- Motivated by the interest of reviewers t3GQ and fNGy in Rukat et al.'s OrM, we now include OrM as an additional competitor in all our experiments, thus also covering the state of the art in probabilistic BMF.
- Motivated by the interest of reviewers Bmfn and t3GQ in the overlap of our synthetic tiles, we now generate overlapping tiles and non-overlapping tiles as part of our evaluation on synthetic data (Sec. 4, Appendix C).

Again, thank you very much for your valuable feedback. Should you be satisfied with the corresponding changes to our submission, we very much hope that you consider reflecting this in your score.

---

### Meta-Review · Area_Chair_G4dW · 2022-08-30

**Recommendation:** Accept
**Confidence:** Less certain

**Metareview:**

This was a border-line paper, but I recommended accepting it , mainly due to the impressive rebuttal that answered at least most of my concerns.
There were issues with the experimental results, but I believe most of them can be fixed since the authors provided full source code (which is not so common).



**Award:**

No

---

### Decision · Program_Chairs · 2022-09-14

Accept